# Initial aortic repair versus medical therapy for early uncomplicated type B dissections

Jyh Shinn Teh[1,2], Jui-Hsiang Chen[1,2], Ying-Ting Kuo[3], Chun-Yang Huang[1,2], Tai-Wei Chen[1,2], Chun-Che Shih[4], Chiao-Po Hsu[1,2,5]*

1 School of Medicine, National Yang Ming Chiao Tung University, Taipei, Taiwan, 2 Division of Cardiovascular Surgery, Department of Surgery, Taipei Veterans General Hospital, Taipei, Taiwan, 3 Department of Radiology, Taipei Veterans General Hospital, Taipei, Taiwan, 4 Division of Cardiovascular Surgery, Department of Surgery, Wan Fang Hospital, Taipei Medical University, Taipei, Taiwan, 5 Department of Surgery, Taoyuan General Hospital, Ministry of Health and Welfare, Taoyuan, Taiwan

* chiaopo@ms39.hinet.net

## Abstract

### Background

Uncomplicated type B aortic dissections was regarded benign and treated with optimal medical therapy (OMT). However, studies showed highly unpredictable of disease progression, which suggested the need of earlier intervention. To search for features associated with worse outcomes with OMT is important due to risk of intervention. We investigated mortality and aortic remodeling between aggressive (OMT and pre-emptive endovascular intervention) and conservative therapy (OMT and necessary operations).

### Methods

Retrospective analysis was performed in acute and subacute uncomplicated type B dissections patients, including typical aortic dissection, intramural hematoma and penetrating atherosclerotic ulcer, diagnosed between June 2005 and May 2021. Patients with Marfan, traumatic, iatrogenic, zone 0 (ascending aorta) involvement, and maximal aortic diameter >45mm were excluded. Patients are classified according to initial management.

### Results

77 and 33 patients in the aggressive and conservative groups were included respectively. There was no differences in baseline clinical and radiological characteristics between them. During mid-term follow-up (median 62.5 months), there was no difference in the mortality but the incidence of 30-day acute kidney injury was significantly higher in aggressive group. Positive aortic remodeling was noted in aggressive group, with development to complete or incomplete false lumen thrombosis (p < 0.01).

### Conclusion

Aggressive pre-emptive endovascular therapy though has acceptable outcomes and positive aortic remodelling in early uncomplicated type B dissection with maximal aortic diameter ≤45mm. However, it could not translate into better mid-term survival than

**Data availability statement:** All relevant data are within the article and its Supporting Information files.

**Funding:** The author(s) received no specific funding for this work.

**Competing interests:** The authors have declared that no competing interests exist.

conservative therapy, but with higher risk of 30-day acute kidney injury. Aggressive pre-emptive endovascular intervention should be cautious in these patients.

## Introduction

Acute aortic syndromes encompass a spectrum of three interrelated life-threatening aortic disease, including typical aortic dissection (TAD), intramural hematoma (IMH) and penetrating atherosclerotic ulcer (PAU). Although pathophysiologic mechanism and evolution differ among these entities, current guidelines suggested similar management strategy in line with management of TAD.[1, 2] Stanford classification system was widely adopted in current practice, which ascending aorta is involved in type A, spared in type B. The wait-and watch strategy with optimal medical therapy (OMT) was suggested for type B aortic syndromes, and surgical interventions with open or endovascular repair (TEVAR) was only considered for complicated cases. However, long-term outcomes of the diseases remain sobering due to aneurysmal expansion of false lumen and other late aortic complications with OMT.[3, 4] Furthermore, the disease progression of acute uncomplicated type B IMH (TBIMH) and PAU (TBPAU) is highly unpredictable, varying from complete absorption, conversion to classic aortic dissection to abrupt aortic rupture.[5, 6]

The INSTEAD (Investigation of STEnt grafts in Acute Dissection) and INSTEAD-XL trial demonstrated reduction of aortic-related mortality and delayed disease progression at 5 years, with pre-emptive TEVAR in 15-day to one-year uncomplicated type B typical aortic dissection (TBTAD).[7, 8] ADSORB trial also tells us pre-emptive TEVAR makes favorable aortic remodeling in one year for acute ($\leq$14 days) uncomplicated TBTAD.[9] In acute uncomplicated TBIMH, Mesar et al, reported IMH thickness > 8mm was an independent risk factor to predictor medical therapy failure and need for endovascular repair,[5] and pre-emptive TEVAR may improve aortic-related adverse events, aortic related mortality and aortic remodeling in patients with high-risk features, such as focal intimal disruption, maximum aortic diameter ($\geq$ 40–45mm) and hematoma thickness ($\geq$ 10mm).[10] Some practitioners have advocated more aggressive treatment with pre-emptive TEVAR.

Due to inherent risk of morbidity and mortality, to search for features that predict worse outcomes with OMT is important for pre-emptive TEVAR. In this study, we aimed to clarify the need to shift toward more aggressive therapeutic approach in uncomplicated type B aortic syndromes with maximal aortic diameter $\leq$ 45mm.

## Methods

### Study population

This retrospective study was approved by the Institutional Review Board of the Taipei Veterans General Hospital (approval number: 2022-02-011CC). Informed written consent to access medical records was obtained from each patient. Selection criteria were patients with acute ($\leq$ 14 days) and subacute (15 to 90 days) uncomplicated type B aortic syndromes, including TAD, and atypical dissection (IMH and PAU), according to SVS/STS classification scheme,[1] and those who received conservative therapy (OMT and necessary TEVAR) or aggressive therapy (OMT and pre-emptive TEVAR) in Taipei Veterans General Hospital from June 2005 to May 2021. Diagnosis of the diseases was established with computed tomography angiography (CTA) examination upon admission. Patients with Marfan syndrome, traumatic or iatrogenic dissections (including during cardiac operation or catheterization) were excluded. Furthermore, the composition is heterogeneous between two groups, to maximize the validity of the

comparison, maximal aortic diameter > 45mm were also excluded in the study. Dissections that involved zone 0 were also excluded due to high uncertainty, as they were regarded as type A dissection in the Stanford classification. Patients are classified according to initial management (conservative or aggressive).

## Outcomes

The clinical follow-up data were collected from records of hospitalization and scheduled monthly clinic evaluations or through direct telephone contact for all-cause (aortic or non-aortic reasons) mortality and major complications, which included all-cause death, cerebral ischemia, spinal cord ischemia, myocardial infarction and aortic rupture. Aorta-related mortality was defined as death from aortic rupture, malperfusion, or aortic dissection, confirmed by sonography or CTA examination.

Clinical examinations and imaging follow-up were performed basically at 1, 3, 6 and 12 months and then annually. Examinations were adjusted in patients with disease progression. Aortic remodeling was assessed with measurement of maximum false and true lumen diameter, hematoma thickness and maximum aortic diameter, as well as thrombosis of false lumen. For follow-up and outcome assessment, a comprehensive, systematic, cross sectional telephone survey was conducted at the endpoint of June 2022.

## Statistical analysis

Data were processed with the SPSS/PC software package version 28. Continuous variables were presented as median (IQR) and compared with (non-parametric statistical) Mann-Whitney U test, while categorical data were compared with Fisher's exact test. Overall survival between two groups was estimated by Kaplan-Meier survival analysis with log-rank test. All tests were 2 tailed, and $P < 0.05$ was considered statistically significant.

## Results

A total of 110 patients were selected from patients with uncomplicated type B aortic syndromes, in acute and subacute setting, in our hospital between June 2005 and May 2021, with follow-up for 62.5 (34.2-101.6) months. (Fig 1) 77 patients received aggressive therapy and the other 33 patients received conservative therapy. The demographic characteristics of all patients are summarized in Table 1, there was no significant differences in clinical and radiographic variables. High risk features of dissection [1] were also compared between them and show no differences (S1 Table).

Procedural details were summarized in S2 Table In the conservative group, the reason of intervention is progressive aneurysm formation. In terms of early outcomes (Table 2), there was no difference in mortality and major complications (cerebral ischemia, spinal cord ischemia, myocardial infarction, and aortic rupture) in both groups, however, aggressive group has significantly longer intensive care unit and hospital stays. The aggressive therapy had higher rate of acute kidney injury (AKI) (16.9% vs. 0%, p = 0.009) than the conservative therapy. 9 patients in the aggressive group had early post-operative endoleak, with 2 patients required re-intervention in one month. Subgroup analysis (TBTAD and atypical TBIMH & TBPAU) did not show unexpected findings (S3 Table and S4).

During longer follow-up, there was no significant differences in all-cause or aorta-related mortality between the two groups, but no doubt more patients with endoleak which may need re-intervention in aggressive therapy. Kaplan-Meier survival analysis revealed similar survival rate between two groups (Fig 2). Subgroup analysis showed similar results (S3 Table and S4).

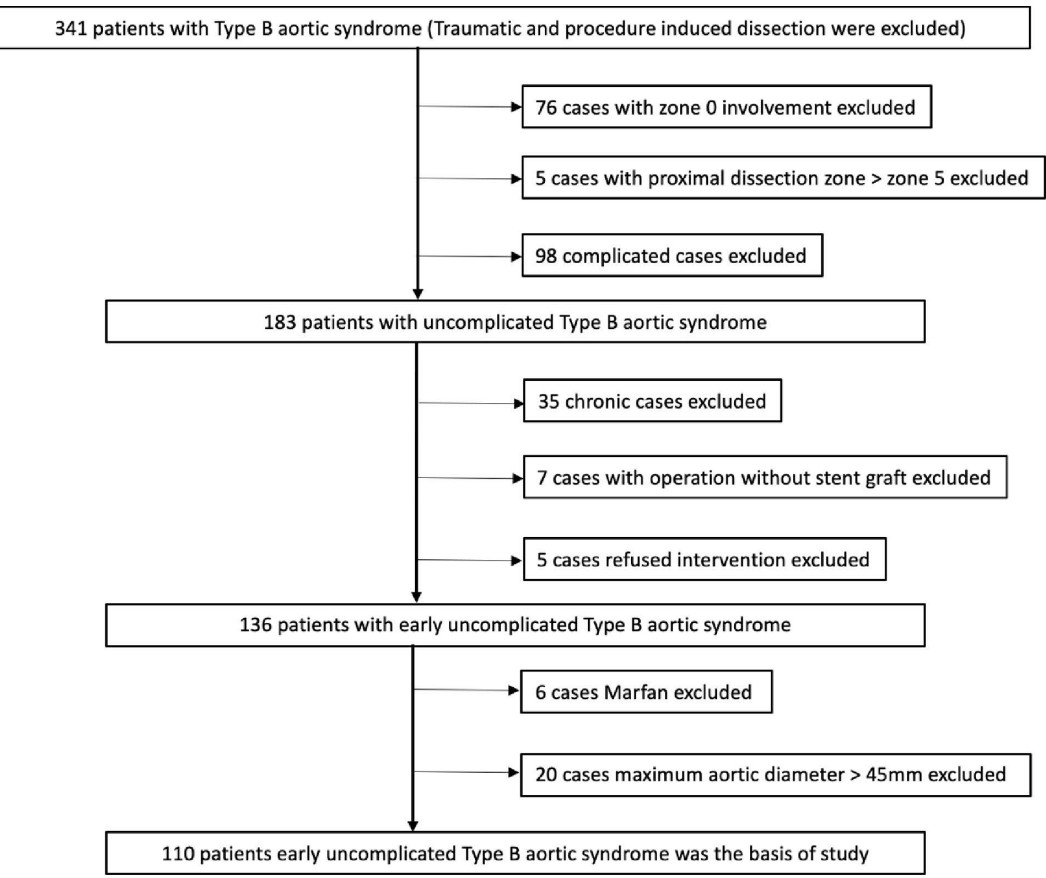

**Fig 1. Study flow diagram.**

Morphological evolution of aorta over time in uncomplicated TBTAD patients was summarized in Table 3. Positive aortic remodeling was found in the aggressive group, but not in conservative group. The maximal false lumen diameter and maximum aortic diameter significantly shrank at 3 months, 1 year and 3 year post stent-grafting. True lumen diameter increased and false lumen diameter decreased at level of or above celiac trunk post stent-grafting (p < 0.005). Aggressive treatment demonstrated significant accelerating false lumen thrombosis within 3-year follow-up (p < 0.01 in Table 2). On the contrary, TBIMH showed positive aortic remodeling over time no matter what aggressive or conservative, and without statistical difference between both groups (Table 4).

## Discussion

In our study, aggressive treatment with early pre-emptive TEVAR failed to demonstrate any survival benefits. However, a higher risk of AKI was identified in patients with a maximum aortic diameter of ≤ 45 mm over a 10-year follow-up period. Pre-emptive TEVAR promoted positive aortic remodeling, evidenced by a reduction in maximum aortic diameter and/or false lumen diameter in type B aortic dissections. However, the positive aortic remodeling did not translate into improved survival outcomes for this group of patients.

Medical therapy is widely accepted as the first-line treatment, and it is a class 1 recommendation for acute uncomplicated type B aortic syndromes.[2,11,12] However, long-term

**Table 1. Baseline characteristics by initial management in uncomplicated type B aortic syndrome.**

| | Aggressive (N = 77) | Conservative (N = 33) | P value |
|---|---|---|---|
| Clinical characteristics | | | |
| Age (y) | 61.0 (51.5– 71.5) | 64.0 (56.5– 78.5) | 0.255 |
| Male | 53 (68.8%) | 22 (66.7%) | 0.823 |
| Hypertension | 56 (72.7%) | 26 (78.8%) | 0.504 |
| DM | 8 (10.4%) | 2 (6.1%) | 0.720 |
| Hyperlipidemia | 2 (2.6%) | 1 (3.0%) | 1 |
| Smoking | 30 (39%) | 9 (27.2%) | 0.148 |
| Cerebrovascular disease | 4 (5.2%) | 0 | 0.314 |
| Coronary artery disease | 8 (10.4%) | 5 (15.2%) | 0.525 |
| Pulmonary disease | 6 (7.8%) | 2 (6.0%) | 1 |
| Previous aortic surgery | 2 (2.6%) | 1 (3.0%) | 1 |
| Uremia | 3 (3.9%) | 1 (3.0%) | 1 |
| eGFR (mL/min/1.73m$^2$)[a] | 68.2 (54.4– 93.9) | 67 (47.8– 75.2) | 0.173 |
| Creatinine (mg/dL)[a] | 1.03 (0.76– 1.26) | 1.08 (0.88– 1.32) | 0.210 |
| Typical/Atypical dissection | 42/35 | 15/18 | 0.382 |
| Acute/Subacute dissection | 64/13 | 29/4 | 0.774 |
| Number of high-risk dissections | 44 (57.1%) | 13 (39.4%) | 0.1 |
| Radiographic characteristics | | | |
| Thoracic aortic diameter (mm) | 36.8 (33.5– 39.2) | 36.3 (33.4– 40.9) | 0.562 |
| Abdominal aortic diameter (mm) | 29.2 (26.9– 32.5) | 28.5 (27.4– 34.3) | 0.809 |
| Maximal aortic diameter (mm) | 37.9 (35.0– 40.0) | 35.6 (33.4– 39.2) | 0.212 |
| Maximal false lumen diameter[b] (mm) | 20 (15.93– 24.73) | 18.2 (13.70– 23.2) | 0.601 |
| Hematoma or ulceration thickness[c] (mm) | 9.9 (7.5– 12) | 8.75 (6.80– 10.6) | 0.053 |
| Proximal dissection level | | | 0.147 |
| Zone 1 | 3 (3.9%) | 1 (3.0%) | |
| Zone 2 | 13 (16.9%) | 2 (6.1%) | |
| Zone 3 | 59 (76.6%) | 27 (81.8%) | |
| Zone 4 | 2 (2.6%) | 1 (3.0%) | |
| Zone 5 | 0 | 2 (6.1%) | |

Pulmonary disease: Chronic obstructive pulmonary disease and/or asthma

[a]calculated with MDRD equation and excluded uremic patients;

[b]only typical dissection;

[c]only atypical dissection.

survival rates continue to be poor.[3] Data from the International Registry of Acute Aortic Dissection (IRAD) have indicated that the short-term outcomes of medical treatment are excellent, whereas the long-term outcomes are poor, with reports of aneurysmal degeneration affecting up to 60% of patients and high follow-up mortality, approaching one in every four patients at 3 years.[3] Several studies have also reported high OMT failure rates in TBIMH cases, with progression to typical aortic dissection, aneurysm formation, and increased hematoma size.[5,13]

ADSORB and INSTEAD, two randomized control trials involving patients with acute (1–14 days) and subacute to chronic (15 days to 1 year) uncomplicated TBTAD, respectively, were not sufficiently powered to demonstrate clinical benefits in terms of short-term (2-year) mortality; however, improvements in aorta-related mortality and disease progression at 5 years were identified in patients randomized to TEVAR.[7–9] Similar results were reported in

**Table 2. Outcomes by initial management in uncomplicated Type B aortic syndrome.**

| | Aggressive (N = 77) | Conservative (N = 33) | P value |
|---|---|---|---|
| **Early outcomes (≦30 days)** | | | |
| Mortality | 1 (1.3%) | 0 | 1 |
| Aortic rupture | 1 (1.3%) | 0 | 1 |
| Acute myocardial infarction | 0 | 0 | |
| Neurological event | 6 (7.8%) | 0 | 0.176 |
| Cerebral ischemia | 4 (5.2%) | 0 | 0.314 |
| Spinal cord | 2 (2.6%) | 0 | 1 |
| Major complications | 6 (7.8%) | 0 | 0.176 |
| Retrograde type A dissection | 1 (1.3%) | | 1 |
| Acute kidney injury[a] | 13 (16.9%) | 0 | 0.009 |
| Post stent-grafting ischemic limb | 1 (1.3%) | X | |
| Post stent-grafting GI bleeding | 1 (1.3%) | X | |
| Post stent-grafting pneumonia | 4 (5.2%) | X | |
| Post stent-grafting respiratory failure | 2 (2.6%) | X | |
| Endoleak | 9 (11.7%) | X | |
| Type 1b | 1 (1.3%) | | |
| Type 2 | 7 (9.1%) | | |
| Type 3 | 1 (1.3%) | | |
| Re-intervention | 2 (2.6%) | X | |
| Intensive care unit stay | 4 (1.5–6) | 2 (0–3) | <0.001 |
| Hospital stay | 16 (11–24.5) | 9 (6.5–13.5) | <0.001 |
| **Cumulative midterm outcomes** | | | |
| All-Cause Mortality | 23 (29.9%) | 12 (36.4%) | 0.503 |
| Aorta-related Mortality | 1 (1.3%) | 0 | 1 |
| Retrograde Type A dissection | 2 (2.6%) | 0 | 1 |
| Endoleak | 15 (19.5%) | 1 (3.0%) | 0.066 |
| Type 1 | 3 (3.9%) | 0 | |
| Type 2 | 11 (13.0%) | 0 | |
| Type 3 | 2 (2.6%) | 1 (3.0%) | |
| Re-intervention | 17 (22.1%) | 2 (6.1%) | 0.042 |
| FL thrombosis within 3 year[b] | N = 38 | N = 14 | <0.001 |
| None/ Incomplete/ Complete | 0/19 (50%)/19 (50%) | 7 (50%)/4 (28.6%)/ 3 (21.4%) | |

Major complication: Cerebral ischemia, spinal cord ischemia, myocardial infarction, and aortic rupture.

[a]increase in serum creatinine of ≧ 0.5 mg/dL or increase to ≧ 150% from baseline.

[b]only typical dissection

other studies.[14, 15] Because of the risks associated with performing a procedure on the vulnerable diseased aorta of asymptomatic patients, pre-emptive TEVAR currently classified as a class 2a or 2b recommendation for acute uncomplicated type B aortic syndromes.[2,11,12]

Therefore, identifying individuals who would benefit from pre-emptive TEVAR might be more helpful. IRAD data and other studies have suggested that long-term adverse events or mortality associated with uncomplicated type B aortic dissections can be predicted by various risk factors, such as maximum aortic diameter, hematoma thickness, false lumen flow, and presence of connective tissue disease.[5,16–19] By targeting these high-risk features, pre-emptive TEVAR could improve survival as early as the first year.[20, 21] Therefore, the "wait and watch" strategy with OMT is not necessarily the optimal or most appropriate care policy

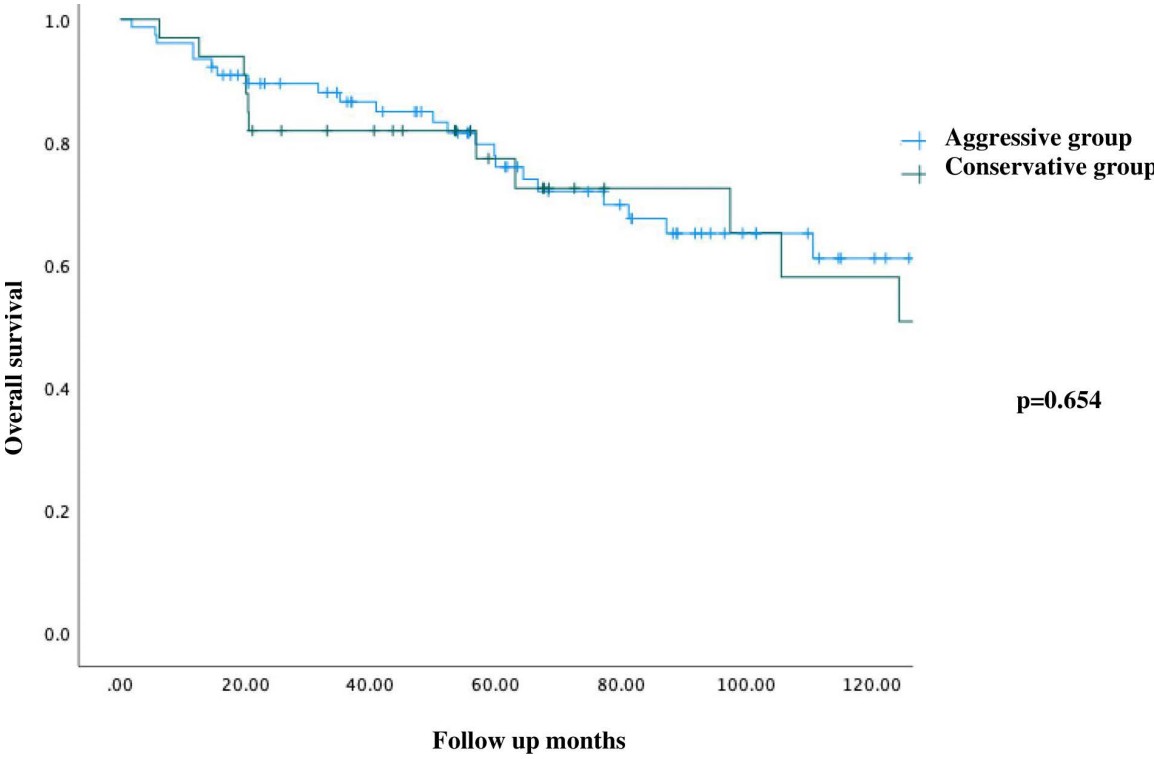

**Fig 2. Kaplan Meier five-year survival of aggressive and conventional groups in uncomplicated dissection.**

for early uncomplicated type B aortic syndrome, and treatment decisions should be based on high-risk factors for mortality or adverse events during follow-up.[10] In our study, we adopted a reverse approach by excluding patients with a maximum aortic diameter of >45mm or Marfan syndrome and following up on these patients. This is because various other risk factors can directly or indirectly affect the aortic diameter. This suggests that the aortic diameter is the central or final marker for these risk factors in determining outcomes. Our study results indicated that pre-emptive TEVAR did not improve clinical benefits even with a follow-up period of 10 years. Furthermore, the 3-year survival exceeded 80% in both groups, which is higher than that reported by Tsai et al.[3] This also suggests that the survival rate is higher for patients with a maximum aortic diameter of ≤45mm than for those with a diameter of >45mm. (S1 Fig). Overall, these findings provide us with valuable insights for clinical practice.

Sailer et al reported a high risk (approximately 25%) of adverse event in the first year after TBIMH onset, with the risk stabilizing during subsequent follow-ups. By contrast, the risk of an adverse event in the first year is approximately 15% for patients with TBTAD and remains consistent in subsequent years.[22] This suggests that the risk of adverse events may differ between TBTAD, TBIMH, and TBPAU. However, no significant difference between these pathologies was identified in our study.

The second issue is the timing of pre-emptive TEVAR. The median time for the aggressive group was 15 days, extending beyond the acute stage in our study. This delay is based on the concept that the intima of the dissected aorta is highly fragile during the acute stage. Numerous studies have suggested that performing TEVAR in the subacute stage yields more favorable results, and more complications, particularly aortic rupture, retrograde type A dissection, and stroke, have been reported in cases of TEVAR conducted during the acute phase (<15 days).[20,23,24]

**Table 3. Aortic remodeling by initial management in uncomplicated typical type B aortic dissection.**

| (Unit: mm) | Aggressive | Conservative | P value |
|---|---|---|---|
| Baseline | N=38 | N=13 | |
| Maximum aortic diameter | 38 (34.75– 40.68) | 35.4 (32.40– 38.50) | 0.121 |
| Maximum FL diameter | 20 (15.93– 24.73) | 18.2 (13.70– 23.20) | 0.337 |
| TL diameter at LSCA level | 18.7 (15.85– 22.75) | 17.2 (10.75– 25.65) | 0.795 |
| FL diameter at LSCA level | 16.9 (14.65– 21.80) | 19.2 (14.00– 24.55) | 0.525 |
| TL diameter at hiatus level | 15.3 (12.50– 20.28) | 16.7 (13.90– 24.45) | 0.266 |
| FL diameter at hiatus level | 16.2 (12.85– 20.85) | 13.7 (10.30– 17.10) | 0.302 |
| TL diameter at celiac trunk level | 13.9 (11.15– 19.88) | 16.4 (12.70– 20.30) | 0.234 |
| FL diameter at celiac trunk level | 13.6 (11.10– 18.40) | 14.0 (10.70– 17.80) | 0.588 |
| TL diameter at renal artery level | 14.45 (11.68– 18.53) | 12.2 (8.93– 20.90) | 0.427 |
| FL diameter at renal artery level | 9.5 (6.53– 14.33) | 13.7 (10.48– 16.33) | 0.123 |
| 3-month remodeling | N=38 | N=12 | |
| Maximum aortic diameter | −3.2 (−5.4– +0.13)[c] | +1.8 (−2.23– +2.85) | 0.012 |
| Maximum FL diameter | −6.2 (−11.73– −2.88)[c] | 0 (−3.08– +3.78) | < 0.001 |
| TL diameter at LSCA level | +11.3 (+6.08– +14.55)[c] | +1.5 (−1.30– +6.15) | < 0.001 |
| FL diameter at LSCA level | −15.7 (−18.5– −9.23)[c] | −0.1 (−9.55– +2.53) | < 0.001 |
| TL diameter at hiatus level | +7 (+3.70– +10.75)[c] | −0.5 (−3.68– +5.85) | 0.001 |
| FL diameter at hiatus level | −10.15 (−16.05– 0)[c] | +1.2 (−3.30– +8.38) | 0.004 |
| TL diameter at celiac trunk level | +3.2 (+0.65– +10.43)[c] | +0.8 (−1.73– +3.90) | 0.022 |
| FL diameter at celiac trunk level | −1.9 (−8.4– +0.03)[c] | +0.7 (−3.5– +7.28) | 0.021 |
| TL diameter at renal artery level | +1.9 (−0.40– +3.60) | −0.6 (−4.50– +2.40) | 0.122 |
| FL diameter at renal artery level | 0 (−2.93– +2.08) | 0 (−1.30– +1.20) | 0.913 |
| 1-Year remodeling | N=32 | N=10 | |
| Maximum aortic diameter | −2.8 (−6.40– −0.20)[c] | +2.1 (−1.90– +7.98) | 0.007 |
| Maximum FL diameter | −8.1 (−12.23– −2.40)[c] | +0.5 (−2.98– +4.45) | 0.008 |
| TL diameter at LSCA level | +10.9 (+7.3– +16.18)[c] | +3.05 (+1.20– +7.80) | < 0.001 |
| FL diameter at LSCA level | −16.4 (−19.40– −12.0)[c] | −1.95 (−11.28– +2.00) | < 0.001 |
| TL diameter at hiatus level | +6.4 (+3.45– +12.80)[c] | −0.5 (−4.33– +6.95) | 0.012 |
| FL diameter at hiatus level | −10.5 (−17.15– −2.1)[c] | +3.95 (−7.43– +10.75) | 0.004 |
| TL diameter at celiac trunk level | + 5.2 (+1.0– +9.53)[c] | +0.2 (−2.75– +1.60) | 0.004 |
| FL diameter at celiac trunk level | −1.5 (−9.38– + 0.78)[c] | +0.3 (−2.43– + 7.18) | 0.053 |
| TL diameter at renal artery level | +1.4 (−1.23– +4.70) | +1.3 (−0.58– +4.88) | 0.953 |
| FL diameter at renal artery level | 0 (−4.35– +2.48)[a] | 0 (−0.43– +1.53) | 0.687 |
| 3-Year remodeling | N=30 | N=9 | |
| Maximum aortic diameter | −2.2 (−6.85– +0.35)[c] | +2.3 (−0.1– +6.45) | 0.006 |
| Maximum FL diameter | −7.55 (−12.35– −3.85)[c] | +0.5 (−5.25– +5.30) | 0.008 |
| TL diameter at LSCA level | +11.25 (+6.58– +16.35)[c] | +6.8 (+1.65– +14.85) | 0.211 |
| FL diameter at LSCA level | −16.35 (−20.13– −11.75)[c] | −12.9 (−15.20– −4.60) | 0.014 |
| TL diameter at hiatus level | +6.3 (+3.15– +10.65)[c] | +4.8 (−1.50– +8.90) | 0.424 |
| FL diameter at hiatus level | −8.7 (−18.20– 0)[c] | −2.4 (−12.95– +10.0) | 0.142 |
| TL diameter at celiac trunk level | +6.0 (+1.30– +11.48)[c] | +1.9 (−0.45– +2.95) | 0.014 |
| FL diameter at celiac trunk level | −3.5 (−13.05– +0.55)[c] | +2.8 (+0.3– +8.15) | 0.015 |
| TL diameter at renal artery level | +2.0 (−0.45– +3.58) | +0.9 (-3.6– +3.15) | 0.301 |
| FL diameter at renal artery level | 0 (−2.68– +5.35) | 0 (−0.35– +1.60) | 0.853 |

FL: false lumen; LSCA: left subclavian artery; TL: true lumen

[a] $p < 0.05$;

[b] $p < 0.01$;

[c] $p < 0.005$ vs. baseline within group.

**Table 4. Aortic remodeling by initial management in uncomplicated Type B IMH.**

| (unit: mm) | Aggressive | Conservative | P value |
|---|---|---|---|
| Baseline | N = 29 | N = 16 | |
| Maximum aortic diameter | 37.05 (35.73– 39.60) | 37.4 (34.60– 39.75) | 0.785 |
| Maximum hematoma thickness | 10 (7.73– 12.08) | 8.75 (6.20– 11.40) | 0.162 |
| 3-month remodeling | N = 25 | N = 8 | |
| Maximum aortic diameter | −4.45 (−6.88– −2.65)[c] | −7.05 (−9.98– −1.75)[a] | 0.396 |
| Maximum hematoma thickness | −5.7 (−8.70– −2.15)[c] | −4.9 (−8.38– −0.53) | 0.557 |
| 1-year remodeling | N = 20 | N = 9 | |
| Maximum aortic diameter | −4.9 (−1.50– −6.43)[c] | −6.6 (−9.05– −2.15) | 0.289 |
| Maximum hematoma thickness | −7.35 (−10.20– −4.68)[c] | −4 (−8.15– −1.85) | 0.045 |
| 3–year remodeling | N = 18 | N = 7 | |
| Maximum aortic diameter | −4.1 (−5.98– −2.85)[c] | −3.8 (−8.90– −2.50)[b] | 0.856 |
| Maximum hematoma thickness | −7.9 (−12.18– −4.78)[c] | −4.9 (−9.10– −2.90)[c] | 0.318 |

[a] $p < 0.05$;

[b] $p < 0.01$;

[c] $p < 0.005$ vs. baseline within group.

The 30-day mortality appears to be better in our study than in previous ones (0 in the OMT group and 1.3% in the TEVAR group of our study vs. 1.5% in the OMT group and 2.8% in the TEVAR group of the INSTEAD trial and 7.6% in the medical group and 8.7% in the TEVAR group in a recently published meta-analysis by Wang et al.).[7,25] However, if patients with Marfan syndrome and a larger aortic diameter (maximum aortic diameter > 45 mm) were included, 30-day mortality would increase to more than 2%. This indirectly supports the hypothesis that the maximum aortic diameter is a significant factor influencing early mortality.[26] However, 30-day complication rate in our study is only generally consistent with those reported in previous studies, without being significantly lower.[7,24] This may be attributed to the inherent risks of TEVAR, even in patients with smaller maximum aortic diameters. Post-operative AKI was significantly higher in the aggressive group in our cohort. The incidence of AKI, which is seldom emphasized, was significantly higher (16.9%) in patients undergoing TEVAR.[27] This complication can be attributed to the use of contrast during stent-graft deployment, which places patients at risk for AKI.

In addition to the higher risk of AKI and stent-graft-related endoleak, longer ICU and hospital stays contribute to increased health-care expenditure associated with hospitalization and the increasing number of invasive procedures. Although we did not assess the cost-effectiveness of these two therapeutic options, from the viewpoint of clinical outcomes, pre-emptive TEVAR may not be an appropriate option for these patients.

Positive aortic remodeling in TBTAD was identified in the aggressive group, with decreases in both the maximum false lumen diameter and the total aortic diameter during the follow-up periods at 3 months, 1 year, and 3 years after stent-grafting. The present study confirmed previous findings that aortic remodeling and false lumen thrombosis after TEVAR is a continuous process with greater remodeling in proximal aortic segments.[8, 9] The VIRTUE registry revealed that most of the aortic remodeling process was complete at 1 year, whereas false-lumen thrombosis rates continued to increase over the course of 3 years.[28] The results for aortic remodeling in the TBIMH did not indicate any obvious beneficial effect in the aggressive group compared with the conservative group. The lack of significance may be associated with the smaller wall stress in patients with smaller aortic diameters, which may facilitate positive aortic remodeling even without TEVAR. Other studies have reported similar observations.[19,29]

Our study demonstrated that OMT is sufficient for patients with acute and subacute uncomplicated type B aortic syndrome with a maximum aortic diameter of ≤ 45 mm, and no significant difference in outcomes after intervention was identified between TBTAD and IMH. Thus, more randomized control studies are warranted to confirm the nonsignificant effects of pre-emptive TEVAR in these patients in the future.

## Conclusion

Aggressive therapy with pre-emptive TEVAR produced acceptable outcomes. However, compared with conservative therapy (OMT), TEVAR did not translate into a survival benefit based on positive aortic remodeling in patients with acute and subacute uncomplicated type B aortic dissection and a maximum aortic diameter of ≤ 45 mm. Although the incidence of major complications, including paraplegia, stroke, and retrograde type A dissection, was low, the 20% incidence of AKI following TEVAR warrants increased attention. Pre-emptive TEVAR should be approached with caution in these patients.

## Limitation of study

There are several limitations of this study. This was a retrospective study during a span of fifteen years, which leads to heterogeneity in various aspects, including management strategy and surgical techniques. Only 30% patients were followed up more than 10 years. Selection bias is possible as the patient grouping was not randomised. Sample size in our study was relatively small and subgroup analysis between acute and subacute disease was not performed in our study. Besides that, optimal therapeutic efficacy of OMT was failed to assess as unavailable information of medication adherence in each patient and exact cause of some mortalities cannot be acquired merely by telephone survey.

## Supporting information

**S1. Table.  High risk features by initial management in uncomplicated type B aortic syndrome.**
(DOCX)

**S2 Table.  Procedural characteristics by initial management in uncomplicated type B aortic syndrome.**
(DOCX)

**S3 Table.  Outcomes by initial management in uncomplicated typical type B aortic dissection.**
(DOCX)

**S4 Table.  Outcomes by initial management in uncomplicated atypical type B aortic dissection.**
(DOCX)

**S1 Fig.  KM Plot of aggressive group (Excluded cases (Marfan, Maximum aortic diameter > 45mm patient) vs included cases) .**
(TIF)

**S1 Data.  Pre-TEVAR baseline characteristic of patients.**
(XLSX)

**S2 Data.  Procedural details, early and late outcome.**
(XLSX)

## Author contributions

**Conceptualization:** Chun-Che Shih, Chiao-Po Hsu.

**Data curation:** Jyh Shinn Teh, Jui-Hsiang Chen, Ying-Ting Kuo.

**Formal analysis:** Jyh Shinn Teh.

**Methodology:** Chun-Yang Huang.

**Supervision:** Chiao-Po Hsu.

**Validation:** Chiao-Po Hsu.

**Writing – original draft:** Jyh Shinn Teh.

**Writing – review & editing:** Chun-Yang Huang, Tai-Wei Chen, Chiao-Po Hsu.

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
