## [Decision Letter · Decision Letter 0]

29 Aug 2024

PONE-D-23-19689Initial aortic repair versus medical therapy for early uncomplicated type B dissectionsPLOS ONE

Dear Dr. Hsu,

Thank you for submitting your manuscript to PLOS ONE. After careful consideration, we feel that it has merit but does not fully meet PLOS ONE’s publication criteria as it currently stands. Therefore, we invite you to submit a revised version of the manuscript that addresses the points raised during the review process.

We look forward to receiving your revised manuscript.

Kind regards,

Eyüp Serhat Çalık

Academic Editor

PLOS ONE

2. We note that your Data Availability Statement is currently as follows: [The data underlying the results presented in the study are available from supporting EXCEL file.]

Additional Editor Comments:

Dear Authors

Optimal medical treatment of uncomplicated Type B aortic dissections has been considered the classical approach, but with the widespread use of TEVAR, immediate interventional or hybrid treatment approaches have become more common. Demonstrating the advantages and disadvantages of these two approaches makes your manuscript important. The manuscript has been evaluated by two reviewers and their recommendations are as follows. I would especially like to emphasize the revision of your statistical methods and language editing. Good luck.

Reviewers' comments:

Reviewer's Responses to Questions

**Comments to the Author**

1. Is the manuscript technically sound, and do the data support the conclusions?

Reviewer #1: No

Reviewer #2: Partly

2. Has the statistical analysis been performed appropriately and rigorously? 

Reviewer #1: No

Reviewer #2: No

3. Have the authors made all data underlying the findings in their manuscript fully available?

Reviewer #1: Yes

Reviewer #2: Yes

4. Is the manuscript presented in an intelligible fashion and written in standard English?

Reviewer #1: Yes

Reviewer #2: Yes

5. Review Comments to the Author

Reviewer #1: With interest, I have read a paper by Jyh Shinn Teh and associates devoted to a topical problem of choosing between medical and interventional treatment of early uncomplicated type B aortic dissection.

There is a big amount of data concerning the recommendation of medical therapy as preferable treatment in early acute uncomplicated Stanford type B aortic dissection. At the same time, some research studies suggest early aggressive approach in treatment of such patients, mostly in the form of transcatheter intervention. The authors performed retrospective comparative analysis of results in patients with acute and subacute uncomplicated type B typical aortic dissection, intramural hematoma and penetrating atherosclerotic aortic ulcer, who received medical and interventional treatment. I would recommend shrinking the Introduction section, and moving the information from it to the Discussion section. The purpose of the study has not been defined. No data about amount of patients with retrograde aortic dissection is provided. A significantly higher proportion of patients with signs of unfavourable prognosis in the interventional treatment group is noteworthy. This may have influenced the results obtained. To offset this effect, I would recommend averaging the patients according to the presence or absence of signs of unfavourable prognosis, all together or individually (maximum aortic diameter 22 mm or more, pleural effusion, refractory hypertension, etc.). I would also recommend excluding patients with Marfan syndrome from the analysis, as there are few of them (3 in each group) and removing them would allow for more "clean" groups and more reliable results. In addition, there is a question of whether transcatheter interventions in patients with Marfan syndrome and acute aortic syndrome are consistent with the recommendation to prefer open surgery in these patients. The question arises as to why the median time from the onset of dissection to surgery in the intervention group was 16 days (from 9 to 26 days) and how correct it is in this situation to talk about treating acute dissection. I would like to know what were the indications for total arch prosthetics under CPB and if the frozen or conventional elephant trunk technique was used. The conclusion of the worst results of aggressive treatment with early TEVAR is based on the incorrect comparison of two different groups. I would recommend performing “cleaning” and averaging of the study groups as mentioned above. To maximize the validity of the comparison, I would recommend that the authors discuss the possibility of performing a propensity score matching. The absence of differences between the results of different types of treatment indicates a correct selection of patients for medical, interventional and open surgical treatment, and the authors' task is to statistically process the data in such a way as to offset the effects of heterogeneous composition of the groups.

I recommend performing English proofreading and major revision.

Reviewer #2: 1. To be distribution-free and no need to account for normal distribution, please use non-parametric statistical methods for all analyses (your sample sizes were also not large), i.e. median+-IQR, Mann-Whitney U test (not t-test), and Fisher's exact test.

2. The presented tables are too complicated and please consolidate, for example, Tables 6 and 7. There were too many measurements. Possible to simplify them and just show aortic remodeling % or Yes/No?

3. Plos ONE is a general-purpose journal, not a cardiovascular or aortic surgery journal. Please rewrite the whole manuscript to make it more reader-friendly to non-specialist audience. For example, please define "aortic remodeling" and explain its relationships to all your measurements in Tables 6 and 7.

4. Please define "survival" or "mortality". Hospital survival? Overall survival? 30-day survival? Hospital mortality? All-cause mortality? Aorta-related mortality? Surgical mortality? 30-day mortality? Please define your hard and soft outcomes. Short-term? Long-term? How to define short- and long-term?

5. You said aggressive group had no survival advantage. Was it long-term, short-term, or overall? Then you said aortic remodeling in the aggressive group may translate into improved long-term survival. How and why? Were they contradicted viewpoints? Please explain with your data by a reader-friendly way.

6. An analog to your topic is "Initial CABG versus PTCA for early uncomplicated CAD-1VD". How do you think the data will show? CABG may show better long-term graft patency but was it worthy the surgical risks and would the patient/family accept it? Over-treatment may show superior efficacy but how about the risk, cost, or life-quality effect? Please discuss this kind of arguments in your discussion.

Simply put, non-parametric statistics, simply your tables, better definitions, discuss over-treatment, and rewrite for non-specialist readers.

6. PLOS authors have the option to publish the peer review history of their article (what does this mean? ). If published, this will include your full peer review and any attached files.

**Do you want your identity to be public for this peer review?** For information about this choice, including consent withdrawal, please see our Privacy Policy .

Reviewer #1: **Yes: ** Vladimir Uspenskiy

Reviewer #2: **Yes: ** Robert J. Chen, MD, MPH

---

## [Author Response · Author response to Decision Letter 0]

28 Oct 2024

PLOS ONE, EDITORIAL OFFICE

27 Oct 2024

Dear Reviewers or Editors,

The feedback provided by reviewers has been helpful to improving this manuscript and we are grateful for their input. We have also responded in detail to all the comments made in the following pages. This rebuttal follows the format in which the points from the reviewers were left in the original order, in blue, and our responses (in black) were then inserted after each of those points.

Reviewer #1

1. I would recommend shrinking the Introduction section, and moving the information from it to the Discussion section.

Authors: According to the reviewer’s suggestion, we have re-written the introduction and discussion sections.

2. The purpose of the study has not been defined.

Authors: We have added in last paragraph of Introduction. (Page 6, Line 102-104)

3. No data about amount of patients with retrograde aortic dissection is provided.

Authors: In Figure 1, 76 patients were excluded because of zone 0 involvement, and they were also known as retrograde aortic dissection cases (in the new reporting standards as B0,x).1

4. A significantly higher proportion of patients with signs of unfavourable prognosis in the interventional treatment group is noteworthy. This may have influenced the results obtained. To offset this effect, I would recommend averaging the patients according to the presence or absence of signs of unfavourable prognosis, all together or individually (maximum aortic diameter 22 mm or more, pleural effusion, refractory hypertension, etc.).

Authors: We would discuss how to manage this issue in answer to comment 9.

5. I would also recommend excluding patients with Marfan syndrome from the analysis, as there are few of them (3 in each group) and removing them would allow for more "clean" groups and more reliable results.

Authors: Marfan syndrome patients were excluded in revised version.

6. In addition, there is a question of whether transcatheter interventions in patients with Marfan syndrome and acute aortic syndrome are consistent with the recommendation to prefer open surgery in these patients.

Authors: Yes, transcatheter intervention was not recommended in Marfan patients. However, this is a retrospective study and some patients without typical characteristics were diagnosed with Marfan syndrome after operation by genetic test. It is not an issue in the revised manuscript as Marfan patients were excluded.

7. The question arises as to why the median time from the onset of dissection to surgery in the intervention group was 16 days (from 9 to 26 days) and how correct it is in this situation to talk about treating acute dissection.

Authors: We use “early” in the topic of this manuscript, which represented “acute and subacute” dissection. These patients are uncomplicated dissection cases and TEVAR could be done non-urgently, and beyond the acute phase. Furthermore, many studies suggested that performing TEVAR in subacute stage would be better. 2-4

8. I would like to know what were the indications for total arch prosthetics under CPB and if the frozen or conventional elephant trunk technique was used.

Authors: This is a retrospective study, and the indication of total arch replacement is not clearly mentioned in medical records. The decision to perform total arch replacement depends on surgeon in-charge, which usually associated with unhealthy aortic arch, arch anatomy or larger aortic arch diameter. Frozen elephant trunk was used in all total arch replacement cases of this study.

9. The conclusion of the worst results of aggressive treatment with early TEVAR is based on the incorrect comparison of two different groups. I would recommend performing “cleaning” and averaging of the study groups as mentioned above. To maximize the validity of the comparison, I would recommend that the authors discuss the possibility of performing a propensity score matching. The absence of differences between the results of different types of treatment indicates a correct selection of patients for medical, interventional and open surgical treatment, and the authors' task is to statistically process the data in such a way as to offset the effects of heterogeneous composition of the groups.

Authors: We tried to offset the effects of heterogeneous composition between these two groups. The association of survival and high-risk features were analyzed, and only “maximal aortic diameter” was strongly related to survival, and the others (ulceration thickness, radiographic malperfusion and IMH with ulcer-like projection) were not. Then distribution of the variable (maximal aortic diameter) was checked in two groups and found that more patients in the aggressive group had larger aortic diameters. Directly excluding patients whose maximal aortic diameter larger than 45mm can correct the pre-treatment difference of variables between the two groups without excluding too many cases (as performing Propensity Score Matching). Hence, we excluded patients with “maximal aortic diameter” larger than 45 mm in the revised manuscript.

10. I recommend performing English proofreading and major revision.

Authors: English proofreading was done.

Reviewer #2:

1. To be distribution-free and no need to account for normal distribution, please use non-parametric statistical methods for all analyses (your sample sizes were also not large), i.e. median+-IQR, Mann-Whitney U test (not t-test), and Fisher's exact test.

Answer: Statistical methods were corrected in the revised manuscript.

2. The presented tables are too complicated and please consolidate, for example, Tables 6 and 7. There were too many measurements. Possible to simplify them and just show aortic remodeling % or Yes/No?

Answer: Most tables and measurements have been simplified. However, details of aortic remodeling were kept as it demonstrated effect of TEVAR over different segment of aorta.

3. Plos ONE is a general-purpose journal, not a cardiovascular or aortic surgery journal. Please rewrite the whole manuscript to make it more reader-friendly to non-specialist audience. For example, please define "aortic remodeling" and explain its relationships to all your measurements in Tables 6 and 7.

Answer: Definition and evaluation of aortic remodeling was mentioned in the method section.

4. Please define "survival" or "mortality". Hospital survival? Overall survival? 30-day survival? Hospital mortality? All-cause mortality? Aorta-related mortality? Surgical mortality? 30-day mortality? Please define your hard and soft outcomes. Short-term? Long-term? How to define short- and long-term?

Answer: We have simplified measurements to 30-day, overall and aortic-related mortality. Definitions were mentioned in Method section. Outcomes were also simplified to short term (30 day) and mid-term (10 years), with major aorta related complications such as aorta rupture and aorta related mortality, procedural complications, neurological complications and else.

5. You said aggressive group had no survival advantage. Was it long-term, short-term, or overall? Then you said aortic remodeling in the aggressive group may translate into improved long-term survival. How and why? Were they contradicted viewpoints? Please explain with your data by a reader-friendly way.

Answer: In the revised manuscript, we focused on patients whose maximum aortic diameter ≤ 45mm. However, in this specific group, aortic remodeling in the aggressive group could not translate into improved mid-term survival.

6. An analog to your topic is "Initial CABG versus PTCA for early uncomplicated CAD-1VD". How do you think the data will show? CABG may show better long-term graft patency but was it worthy the surgical risks and would the patient/family accept it? Over-treatment may show superior efficacy but how about the risk, cost, or life-quality effect? Please discuss this kind of arguments in your discussion.

Answer: A short paragraph mentioned about cost-effectiveness was added in the discussion section of revised manuscript.

Reference:

1.Lombardi JV, Hughes GC, Appoo JJ, Bavaria JE, Beck AW, Cambria RP, Charlton-Ouw K, Eslami MH, Kim KM, Leshnower BG, Maldonado T, Reece TB and Wang GJ. Society for Vascular Surgery (SVS) and Society of Thoracic Surgeons (STS) Reporting Standards for Type B Aortic Dissections. The Annals of thoracic surgery. 2020;109:959-981.

2.Potter HA, Ding L, Han SM, Weaver FA, Beck AW, Malas MB and Magee GA. Impact of high-risk features and timing of repair for acute type B aortic dissections. Journal of vascular surgery. 2022;76:364-371.e3.

3.Jubouri M, Al-Tawil M, Yip HCA, Bashir A, Tan S, Bashir M, Anderson R, Bailey D, Nienaber CA, Coselli JS and Williams I. Mid- and long-term outcomes of thoracic endovascular aortic repair in acute and subacute uncomplicated type B aortic dissection. Journal of cardiac surgery. 2022;37:1328-1339.

4.Xie E, Yang F, Liu Y, Xue L, Fan R, Xie N, Chen L, Liu J and Luo J. Timing and Outcome of Endovascular Repair for Uncomplicated Type B Aortic Dissection. European journal of vascular and endovascular surgery : the official journal of the European Society for Vascular Surgery. 2021;61:788-797.

---

## [Decision Letter · Decision Letter 1]

3 Dec 2024

PONE-D-23-19689R1Initial aortic repair versus medical therapy for early uncomplicated type B dissectionsPLOS ONE

Dear Dr. Hsu,

Thank you for submitting your manuscript to PLOS ONE. After careful consideration, we feel that it has merit but does not fully meet PLOS ONE’s publication criteria as it currently stands. Therefore, we invite you to submit a revised version of the manuscript that addresses the points raised during the review process.

We look forward to receiving your revised manuscript.

Kind regards,

Eyüp Serhat Çalık

Academic Editor

PLOS ONE

Additional Editor Comments:

The manuscript was evaluated by one previous and one new reviewer. Their suggestions are below. Please upload your manuscript as soon as possible with the revisions you will make along with your point-by-point responses. Good luck.

Reviewers' comments:

Reviewer's Responses to Questions

**Comments to the Author**

1. If the authors have adequately addressed your comments raised in a previous round of review and you feel that this manuscript is now acceptable for publication, you may indicate that here to bypass the “Comments to the Author” section, enter your conflict of interest statement in the “Confidential to Editor” section, and submit your "Accept" recommendation.

Reviewer #2: All comments have been addressed

Reviewer #3: (No Response)

2. Is the manuscript technically sound, and do the data support the conclusions?

Reviewer #2: Yes

Reviewer #3: Yes

3. Has the statistical analysis been performed appropriately and rigorously? 

Reviewer #2: Yes

Reviewer #3: Yes

4. Have the authors made all data underlying the findings in their manuscript fully available?

Reviewer #2: Yes

Reviewer #3: No

5. Is the manuscript presented in an intelligible fashion and written in standard English?

Reviewer #2: Yes

Reviewer #3: Yes

6. Review Comments to the Author

Reviewer #2: After carefully reviewing the revised manuscript and the authors' responses, I am pleased with the improvements made based on previous feedback. The authors effectively addressed critical points from my initial review, specifically the implementation of non-parametric statistical methods and the simplification of tables, which enhances the manuscript’s readability. Additionally, the clarification of complex terms, such as "aortic remodeling," and the distinctions between short- and long-term outcomes improve accessibility for the PLOS ONE readership. The nuanced discussion about aggressive management benefits, including aortic remodeling without survival benefits, is balanced and well-articulated. I commend the authors for thoroughly integrating these modifications, which significantly strengthen the clarity and relevance of the findings.

Reviewer #3: 1. If you exclude the maximum aortic diameter >45mm, you should discuss about these patients separately since these are the high risk patients.

2. Original conclusion is better than the revised conclusion. Revised conclusion sounds very negative.

3. I don't see the procedure details which is Table S2...

4. Some patients in both groups have zone 1,2 involvement. These are type A dissection, not type B. So I am not sure you should include these patients.

5. To be honest, I don't know what's new on this paper...

7. PLOS authors have the option to publish the peer review history of their article (what does this mean? ). If published, this will include your full peer review and any attached files.

**Do you want your identity to be public for this peer review?** For information about this choice, including consent withdrawal, please see our Privacy Policy .

Reviewer #2: **Yes: ** Robert J. Chen, MD, MPH

Reviewer #3: No

---

## [Author Response · Author response to Decision Letter 1]

21 Dec 2024

PLOS ONE, EDITORIAL OFFICE

15 Dec 2024

Dear Reviewers or Editors,

The feedback provided by reviewers has been helpful to improving this manuscript and we are grateful for their input. We have responded in detail to all the comments by reviewer #3 made in the following pages.

Reviewer #3:

1. If you exclude the maximum aortic diameter >45mm, you should discuss about these patients separately since these are the high risk patients.

Figure 1: KM Plot of excluded cases (Marfan, Maximum aortic diameter > 45mm patient) vs. included cases

Figure 2: KM Plot of aggressive group (Excluded cases vs included cases)

From Figure 1 and 2, patient with Marfan syndrome and maximum aortic diameter > 45mm had worse survival outcome than included cases, with statistically significant. This indirectly supports the hypothesis that the maximum aortic diameter is a significant factor influencing early mortality.

There were 3 cases (7.9%) with Marfan or maximum aortic diameter >45mm in the conservative group. On the contrary, there were 18 cases (18.4%) with Marfan or maximum aortic diameter >45mm in the aggressive group. Hence, if we did not exclude these patients, the power of comparison may be not enough. Furthermore, most patients whose maximum aortic diameter >45mm received pre-emptive TEVAR. In consideration of these factors, we excluded these cases in our study at last. The above Figure 2 was added in the discussion section of revised manuscript.

2. Original conclusion is better than the revised conclusion. Revised conclusion sounds very negative.

The revised conclusion is no survival benefits with aggressive treatment, but positive remodeling of aorta in type B patients with maximal aortic diameter ≤45mm. (Focused on the group maximal aortic diameter ≤45mm)

3. I don't see the procedure details which is Table S2...

Procedural details were showed in the Table S2, which could be downloaded via the link of supplemental tables in the page 40 of manuscript PDF file.

4. Some patients in both groups have zone 1,2 involvement. These are type A dissection, not type B. So I am not sure you should include these patients.

Although newer classification system of aortic dissection has been proposed,1 we used the SVS/STS classification scheme for aortic dissection proposed in 2020.2 In this classification, the arch involved (zone 1, 2) type B dissection was not regarded as type A. In our study, we used this classification scheme to collect patients and written in the method section.

5. To be honest, I don't know what's new on this paper

In our study, pre-emptive TEVAR promotes positive aortic remodeling in type B patients. However, no survival benefits were noted in patients with maximal aortic diameter ≤45mm compared with optimal medical control. Moreover, higher risk of 30-day acute kidney injury was noted after pre-emptive TEVAR. It hints that the decision should be more careful to balance the risk of TEVAR with the benefit of aortic remodeling only in this specific group of patients.

Reference:

1. Ramesh P, Al-Zubaidi FI, Abdelghaffar M, Babiker S, Aspinall A, Butt S, Sabry H, Zeinah M and Harky A. TEM Classification of Aortic Dissection-The Evolving Scoring System: A Literature Review. Heart, lung & circulation. 2024;33:17-22.

2. Lombardi JV, Hughes GC, Appoo JJ, Bavaria JE, Beck AW, Cambria RP, Charlton-Ouw K, Eslami MH, Kim KM, Leshnower BG, Maldonado T, Reece TB and Wang GJ. Society for Vascular Surgery (SVS) and Society of Thoracic Surgeons (STS) Reporting Standards for Type B Aortic Dissections. The Annals of thoracic surgery. 2020;109:959-981.

** For more details, please see Response to reviewers

---

## [Decision Letter · Decision Letter 2]

7 Jan 2025

PONE-D-23-19689R2Initial aortic repair versus medical therapy for early uncomplicated type B dissectionsPLOS ONE

Dear Dr. Hsu,

Thank you for submitting your manuscript to PLOS ONE. After careful consideration, we feel that it has merit but does not fully meet PLOS ONE’s publication criteria as it currently stands. Therefore, we invite you to submit a revised version of the manuscript that addresses the points raised during the review process.

We look forward to receiving your revised manuscript.

Kind regards,

Eyüp Serhat Çalık

Academic Editor

PLOS ONE

Journal Requirements:

Additional Editor Comments:

I would like to thank the authors for their appropriate revisions and point-by-point responses. Your manuscript has been reviewed by previous reviewers and here are some of their suggestions for minor corrections. Good luck.

Note: Please note the attached pages.

Reviewers' comments:

Reviewer's Responses to Questions

**Comments to the Author**

1. If the authors have adequately addressed your comments raised in a previous round of review and you feel that this manuscript is now acceptable for publication, you may indicate that here to bypass the “Comments to the Author” section, enter your conflict of interest statement in the “Confidential to Editor” section, and submit your "Accept" recommendation.

Reviewer #1: All comments have been addressed

Reviewer #2: All comments have been addressed

Reviewer #3: (No Response)

2. Is the manuscript technically sound, and do the data support the conclusions?

Reviewer #1: Yes

Reviewer #2: Yes

Reviewer #3: Yes

3. Has the statistical analysis been performed appropriately and rigorously? 

Reviewer #1: Yes

Reviewer #2: Yes

Reviewer #3: Yes

4. Have the authors made all data underlying the findings in their manuscript fully available?

Reviewer #1: Yes

Reviewer #2: Yes

Reviewer #3: Yes

5. Is the manuscript presented in an intelligible fashion and written in standard English?

Reviewer #1: No

Reviewer #2: Yes

Reviewer #3: Yes

6. Review Comments to the Author

Reviewer #1: The list of references used needs to be updated. Of the 32 papers used, 24 (75%) were published earlier than 2019 (more than 5 years ago), including 5 papers published in 2004 and earlier (20 or more years ago).

The text contains a number of typographical errors (e.g., on line 82) as well as some syntylic errors. I recommend performing English proofreading by a native speaker.

In general, the paper is technically sound, and the methods are appropriate and properly conducted. The results and conclusions logically follow from the conducted analysis. The data are adequately interpreted. The statistical analysis is sound. The claims are fully supported by the study data. There are no special ethical concerns regarding the use of human subjects. I have no other special comments on the work, and I recommend it for publication after minor revision.

Thank you for submitting your study to the Journal and good luck for the paper.

Reviewer #2: The authors have addressed the reviewer's comments thoroughly and thoughtfully. They provided additional analyses and clarifications to support their methodology and findings. The revised conclusion appropriately reflects the study's results, highlighting both the positive aortic remodeling and lack of survival benefits in the specific patient group. The manuscript's novelty lies in its focus on patients with maximal aortic diameter ≤45mm, offering valuable insights for clinical decision-making in this subset.

Reviewer #3: Still you didn't answer my question about Figure 1. Figure 1 is study flow diagram, but you mentioned Figure 1 is KM plot of excluded cases in the"Author's Response to Reviewer's Comments". I still need to see the outcome of excluded cases. Same as Figure 2 which you mentioned "KM plot of aggressive group" which I don't see...

7. PLOS authors have the option to publish the peer review history of their article (what does this mean? ). If published, this will include your full peer review and any attached files.

**Do you want your identity to be public for this peer review?** For information about this choice, including consent withdrawal, please see our Privacy Policy .

Reviewer #1: **Yes: ** Vladimir Uspenskiy, MD, DrHabil, Associate Professor

Reviewer #2: **Yes: ** Robert Jeenchen Chen, MD, MPH

Reviewer #3: No

---

## [Author Response · Author response to Decision Letter 2]

29 Jan 2025

1. Some of the references have been updated with relevant, which showed in the revised manuscript.

2. Both of the figures were attached in “ Response to Reviewers” instead of manuscript. We will redefined them as Figure 3 and Figure 4.

From Figure 3 and 4, patient with Marfan syndrome and maximum aortic diameter > 45mm had worse survival outcome than included cases, with statistically significant. This indirectly supports the hypothesis that the maximum aortic diameter is a significant factor influencing early mortality.

There were 3 cases (7.9%) with Marfan or maximum aortic diameter >45mm in the conservative group. On the contrary, there were 18 cases (18.4%) with Marfan or maximum aortic diameter >45mm in the aggressive group. Hence, if we did not exclude these patients, the power of comparison may be not enough. Furthermore, most patients whose maximum aortic diameter >45mm received pre-emptive TEVAR. In consideration of these factors, we excluded these cases in our study at last. The above Figure 4 was added in the discussion section of revised manuscript (showed as Figure S1)..

---

## [Editor Report · Decision Letter 3]

5 Feb 2025

Initial aortic repair versus medical therapy for early uncomplicated type B dissections

PONE-D-23-19689R3

Dear Dr. Hsu,

We’re pleased to inform you that your manuscript has been judged scientifically suitable for publication and will be formally accepted for publication once it meets all outstanding technical requirements.

Kind regards,

Eyüp Serhat Çalık

Academic Editor

PLOS ONE
---

## [Editor Report · Acceptance letter]

PONE-D-23-19689R3

PLOS ONE

Dear Dr. Hsu,

I'm pleased to inform you that your manuscript has been deemed suitable for publication in PLOS ONE. Congratulations! Your manuscript is now being handed over to our production team.

Kind regards,

on behalf of

Dr. Eyüp Serhat Çalık

Academic Editor

PLOS ONE